# Overcoming Catastrophic Forgetting by Incremental Moment Matching

**Sang-Woo Lee**[1], **Jin-Hwa Kim**[1], **Jaehyun Jun**[1], **Jung-Woo Ha**[2], and **Byoung-Tak Zhang**[1,3]

Seoul National University[1]
Clova AI Research, NAVER Corp[2]
Surromind Robotics[3]

{slee,jhkim,jhjun}@bi.snu.ac.kr  jungwoo.ha@navercorp.com
btzhang@bi.snu.ac.kr

## Abstract

Catastrophic forgetting is a problem of neural networks that loses the information of the first task after training the second task. Here, we propose a method, i.e. incremental moment matching (IMM), to resolve this problem. IMM incrementally matches the moment of the posterior distribution of the neural network which is trained on the first and the second task, respectively. To make the search space of posterior parameter smooth, the IMM procedure is complemented by various transfer learning techniques including weight transfer, L2-norm of the old and the new parameter, and a variant of dropout with the old parameter. We analyze our approach on a variety of datasets including the MNIST, CIFAR-10, Caltech-UCSD-Birds, and Lifelog datasets. The experimental results show that IMM achieves state-of-the-art performance by balancing the information between an old and a new network.

## 1 Introduction

Catastrophic forgetting is a fundamental challenge for artificial general intelligence based on neural networks. The models that use stochastic gradient descent often forget the information of previous tasks after being trained on a new task [1, 2]. Online multi-task learning that handles such problems is described as *continual learning*. This classic problem has resurfaced with the renaissance of deep learning research [3, 4].

Recently, the concept of applying a regularization function to a network trained by the old task to learning a new task has received much attention. This approach can be interpreted as an approximation of sequential Bayesian [5, 6]. Representative examples of this regularization approach include learning without forgetting [7] and elastic weight consolidation [8]. These algorithms succeeded in some experiments where their own assumption of the regularization function fits the problem.

Here, we propose incremental moment matching (IMM) to resolve the catastrophic forgetting problem. IMM uses the framework of Bayesian neural networks, which implies that uncertainty is introduced on the parameters in neural networks, and that the posterior distribution is calculated [9, 10]. The dimension of the random variable in the posterior distribution is the number of the parameters in the neural networks. IMM approximates the mixture of Gaussian posterior with each component representing parameters for a single task to one Gaussian distribution for a combined task. To merge the posteriors, we introduce two novel methods of moment matching. One is *mean-IMM*, which simply averages the parameters of two networks for old and new tasks as the minimization of the average of KL-divergence between one approximated posterior distribution for the combined task

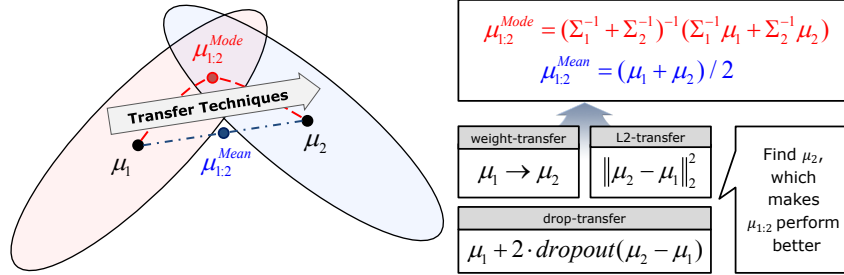

Figure 1: Geometric illustration of incremental moment matching (IMM). Mean-IMM simply averages the parameters of two neural networks, whereas mode-IMM tries to find a maximum of the mixture of Gaussian posteriors. To make IMM be reasonable, the search space of the loss function between the posterior means $\mu_1$ and $\mu_2$ should be reasonably smooth and convex-like. To find a $\mu_2$ which satisfies this condition of a smooth and convex-like path from $\mu_1$, we propose applying various transfer techniques for the IMM procedure.

and each Gaussian posterior for the single task [11]. The other is *mode-IMM*, which merges the parameters of two networks using a Laplacian approximation [9] to approximate a mode of the mixture of two Gaussian posteriors, which represent the parameters of the two networks.

In general, it is too naïve to assume that the final posterior distribution for the whole task is Gaussian. To make our IMM work, the search space of the loss function between the posterior means needs to be smooth and convex-like. In other words, there should not be high cost barriers between the means of the two networks for an old and a new task. To make our assumption of Gaussian distribution for neural network reasonable, we applied three main transfer learning techniques on the IMM procedure: weight transfer, L2-norm of the old and the new parameters, and our newly proposed variant of dropout using the old parameters. The whole procedure of IMM is illustrated in Figure 1.

## 2 Previous Works on Catastrophic Forgetting

One of the major approaches preventing catastrophic forgetting is to use an ensemble of neural networks. When a new task arrives, the algorithm makes a new network, and shares the representation between the tasks [12, 13]. However, this approach has a complexity issue, especially in inference, because the number of networks increases as the number of new tasks that need to be learned increases.

Another approach studies the methods using implicit distributed storage of information, in typical stochastic gradient descent (SGD) learning. These methods use the idea of dropout, maxout, or neural module to distributively store the information for each task by making use of the large capacity of the neural network [4]. Unfortunately, most studies following this approach had limited success and failed to preserve performance on the old task when an extreme change to the environment occurred [3]. Alternatively, Fernando et al. [14] proposed PathNet, which extends the idea of the ensemble approach for parameter reuse [13] within a single network. In PathNet, a neural network has ten or twenty modules in each layer, and three or four modules are picked for one task in each layer by an evolutionary approach. This method alleviates the complexity issue of the ensemble approach to continual learning in a plausible way.

The approach with a regularization term also has received attention. Learning without forgetting (LwF) is one example of this approach, which uses the pseudo-training data from the old task [7]. Before learning the new task, LwF puts the training data of the new task into the old network, and uses the output as pseudo-labels of the pseudo-training data. By optimizing both the pseudo-training data of the old task and the real data of the new task, LwF attempts to prevent catastrophic forgetting. This framework is promising where the properties of the pseudo training set is similar to the ideal training set. Elastic weight consolidation (EWC), another example of this approach, uses sequential Bayesian estimation to update neural networks for continual learning [8]. In EWC, the posterior distribution trained by the previous task is used to update the new prior distribution. This new prior is used for learning the new posterior distribution of the new task in a Bayesian manner.

EWC assumes that the covariance matrix of the posterior is diagonal and there are no correlations between the nodes. Though this assumption is fragile, EWC performs well in some domains.

EWC is a monumental recent work that uses sequential Bayesian for continual learning of neural networks. However, updating the parameter of complex hierarchical models by sequential Bayesian estimation is not new [5]. Sequential Bayes was used to learn topic models from stream data by Broderick et al. [6]. Huang et al. applied sequential Bayesian to adapt a deep neural network to the specific user in the speech recognition domain [15, 16]. They assigned the layer for the user adaptation and applied MAP estimation to this single layer. Similar to our IMM method, Bayesian moment matching is used for sum-product networks, a kind of deep hierarchical probabilistic model [17]. Though sum-product networks are usually not scalable to large datasets, their online learning method is useful, and it achieves similar performance to the batch learner. Our method using moment matching focuses on continual learning and deals with significantly different statistics between tasks, unlike the previous method.

## 3 Incremental Moment Matching

In incremental moment matching (IMM), the moments of posterior distributions are matched in an incremental way. In our work, we use a Gaussian distribution to approximate the posterior distribution of parameters. Given $K$ sequential tasks, we want to find the optimal parameter $\mu^*_{1:K}$ and $\Sigma^*_{1:K}$ of the Gaussian approximation function $q_{1:K}$ from the posterior parameter for each $k$th task, $(\mu_k, \Sigma_k)$.

$$p_{1:K} \equiv p(\theta|X_1, \cdots, X_K, y_1, \cdots, y_K) \approx q_{1:K} \equiv q(\theta|\mu_{1:K}, \Sigma_{1:K}) \tag{1}$$

$$p_k \equiv p(\theta|X_k, y_k) \approx q_k \equiv q(\theta|\mu_k, \Sigma_k) \tag{2}$$

$q_{1:K}$ denotes an approximation of the true posterior distribution $p_{1:K}$ for the whole task, and $q_k$ denotes an approximation of the true posterior distribution $p_k$ over the training dataset $(X_k, y_k)$ for the $k$th task. $\theta$ denotes the vectorized parameter of the neural network. The dimension of $\mu_k$ and $\mu_{1:k}$ is $D$, and the dimension of $\Sigma_k$ and $\Sigma_{1:k}$ is $D \times D$, respectively, where $D$ is the dimension of $\theta$. For example, a multi-layer perceptrons (MLP) with [784-800-800-800-10] has the number of nodes, $D = 1917610$ including bias terms.

Next, we explain two proposed moment matching algorithms for the continual learning of modern deep neural networks. The two algorithms generate two different moments of Gaussian with different objective functions for the same dataset.

### 3.1 Mean-based Incremental Moment Matching (mean-IMM)

**Mean-IMM** averages the parameters of two networks in each layer, using mixing ratios $\alpha_k$ with $\sum_k^K \alpha_k = 1$. The objective function of mean-IMM is to minimize the following *local KL-distance* or the weighted sum of KL-divergence between each $q_k$ and $q_{1:K}$ [11, 18]:

$$\mu^*_{1:K}, \Sigma^*_{1:K} = \underset{\mu_{1:K}, \Sigma_{1:K}}{\operatorname{argmin}} \sum_k^K \alpha_k KL(q_k||q_{1:K}) \tag{3}$$

$$\mu^*_{1:K} = \sum_k^K \alpha_k \mu_k \tag{4}$$

$$\Sigma^*_{1:K} = \sum_k^K \alpha_k (\Sigma_k + (\mu_k - \mu^*_{1:K})(\mu_k - \mu^*_{1:K})^T) \tag{5}$$

$\mu^*_{1:K}$ and $\Sigma^*_{1:K}$ are the optimal solution of the local KL-distance. Notice that covariance information is not needed for mean-IMM, since calculating $\mu^*_{1:K}$ does not require any $\Sigma_k$. A series of $\mu_k$ is sufficient to perform the task. The idea of mean-IMM is commonly used in shallow networks [19, 20]. However, the contribution of this paper is to discover when and how mean-IMM can be applied in modern deep neural networks and to show it can performs better with other transfer techniques.

Future works may include other measures to merge the networks, including the KL-divergence between $q_{1:K}$ and the mixture of each $q_k$ (i.e. $KL(q_{1:K}||\sum_k^K \alpha_k q_k)$) [18].

## 3.2 Mode-based Incremental Moment Matching (mode-IMM)

**Mode-IMM** is a variant of mean-IMM which uses the covariance information of the posterior of Gaussian distribution. In general, a weighted average of two mean vectors of Gaussian distributions is not a mode of MoG. In discriminative learning, the maximum of the distribution is of primary interest. According to Ray and Lindsay [21], all the modes of MoG with $K$ clusters lie on $(K-1)$-dimensional hypersurface $\{\theta|\theta = (\sum_k^K a_k \Sigma_k^{-1})^{-1}(\sum_k^K a_k \Sigma_k^{-1}\mu_k), 0 < a_k < 1 \text{ and } \sum_k a_k = 1\}$. See Appendix A for more details.

Motivated by the above description, a mode-IMM approximate MoG with Laplacian approximation, in which the logarithm of the function is expressed by the Taylor expansion [9]. Using Laplacian approximation, the MoG is approximated as follows:

$$\log q_{1:K} \approx \sum_k^K \alpha_k \log q_k + C = -\frac{1}{2}\theta^T(\sum_k^K \alpha_k \Sigma_k^{-1})\theta + (\sum_k^K \alpha_k \Sigma_k^{-1}\mu_k)\theta + C' \tag{6}$$

$$\mu_{1:K}^* = \Sigma_{1:K}^* \cdot (\sum_k^K \alpha_k \Sigma_k^{-1}\mu_k) \tag{7}$$

$$\Sigma_{1:K}^* = (\sum_k^K \alpha_k \Sigma_k^{-1})^{-1} \tag{8}$$

For Equation 8, we add $\epsilon I$ to the term to be inverted in practice, with an identity matrix $I$ and a small constant $\epsilon$.

Here, we assume diagonal covariance matrices, which means that there is no correlation among parameters. This diagonal assumption is useful, since it decreases the number of parameters for each covariance matrix from $O(D^2)$ to $O(D)$ for the dimension of the parameters $D$.

For covariance, we use the inverse of a Fisher information matrix, following [8, 22]. The main idea of this approximation is that the square of gradients for parameters is a good indicator of their precision, which is the inverse of the variance. The Fisher information matrix for the $k$th task, $F_k$ is defined as:

$$F_k = E\left[\frac{\partial}{\partial \mu_k}\ln p(\tilde{y}|x, \mu_k) \cdot \frac{\partial}{\partial \mu_k}\ln p(\tilde{y}|x, \mu_k)^T\right], \tag{9}$$

where the probability of the expectation follows $x \sim \pi_k$ and $\tilde{y} \sim p(y|x, \mu_k)$, where $\pi_k$ denotes an empirical distribution of $X_k$.

# 4 Transfer Techniques for Incremental Moment Matching

In general, the loss function of neural networks is not convex. Consider that shuffling nodes and their weights in a neural network preserves the original performance. If the parameters of two neural networks initialized independently are averaged, it might perform poorly because of the high cost barriers between the parameters of the two neural networks [23]. However, we will show that various transfer learning techniques can be used to ease this problem, and make the assumption of Gaussian distribution for neural networks reasonable. In this section, we introduce three practical techniques for IMM, including weight-transfer, L2-transfer, and drop-transfer.

## 4.1 Weight-Transfer

**Weight-transfer** initialize the parameters for the new task $\mu_k$ with the parameters of the previous task $\mu_{k-1}$ [24]. In our experiments, the use of weight-transfer was critical to the continual learning performance. For this reason, the experiments on IMM in this paper use the weight-transfer technique by default.

The weight-transfer technique is motivated by the geometrical property of neural networks discovered in the previous work [23]. They found that there is a straight path from the initial point to the solution without any high cost barrier, in various types of neural networks and datasets. This discovery suggests that the weight-transfer from the previous task to the new task makes a smooth loss

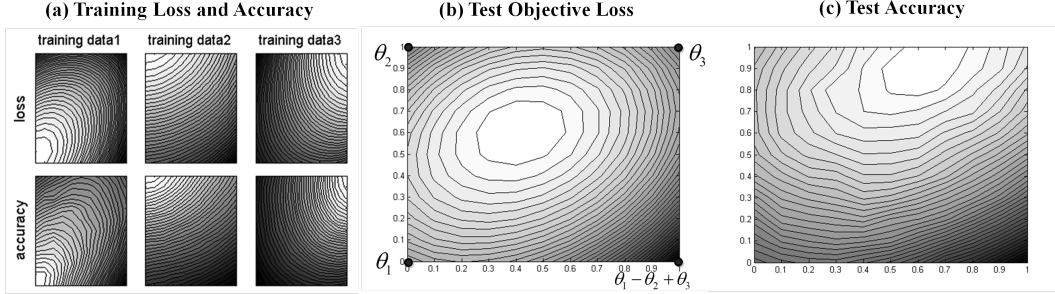

Figure 2: Experimental results on visualizing the effect of weight-transfer. The geometric property of the parameter space of the neural network is analyzed. Brighter is better. $\theta_1$, $\theta_2$, and $\theta_3$ are the vectorized parameters of trained networks from randomly selected subsets of the CIFAR-10 dataset. This figure shows that there are better solutions between the three locally optimized parameters.

surface between two solutions for the tasks, so that the optimal solution for both tasks lies on the interpolated point of the two solutions.

To empirically validate the concept of weight-transfer, we use the linear path analysis proposed by Goodfellow et al. [23] (Figure 2). We randomly chose 18,000 instances from the training dataset of CIFAR-10, and divided them into three subsets of 6,000 instances each. These three subsets are used for sequential training by CNN models, parameterized by $\theta_1$, $\theta_2$, and $\theta_3$, respectively. Here, $\theta_2$ is initialized from $\theta_1$, and then $\theta_3$ is initialized from $\theta_2$, in the same way as weight-transfer. In this analysis, each loss and accuracy is evaluated at a series of points $\theta = \theta_1 + \alpha(\theta_2 - \theta_1) + \beta(\theta_3 - \theta_2)$, varying $\alpha$ and $\beta$. In Figure 2, the loss surface of the model on each online subset is nearly convex. The figure shows that the parameter at $\frac{1}{3}(\theta_1 + \theta_2 + \theta_3)$, which is the same as the solution of mean-IMM, performs better than any other reference points $\theta_1$, $\theta_2$, or $\theta_3$. However, when $\theta_2$ is not initialized by $\theta_1$, the convex-like shape disappears, since there is a high cost barrier between the loss function of $\theta_1$ and $\theta_2$.

## 4.2 L2-transfer

**L2-transfer** is a variant of L2-regularization. L2-transfer can be interpreted as a special case of EWC where the prior distribution is Gaussian with $\lambda I$ as a covariance matrix. In L2-transfer, a regularization term of the distance between $\mu_{k-1}$ and $\mu_k$ is added to the following objective function for finding $\mu_k$, where $\lambda$ is a hyperparameter:

$$\log p(y_k|X_k, \mu_k) - \lambda \cdot ||\mu_k - \mu_{k-1}||_2^2 \tag{10}$$

The concept of L2-transfer is commonly used in transfer learning [25, 26] and continual learning [7, 8] with large $\lambda$. Unlike the previous usage of large $\lambda$, we use small $\lambda$ for the IMM procedure. In other words, $\mu_k$ is first trained by Equation 10 with small $\lambda$, and then merged to $\mu_{1:k}$ in our IMM. Since we want to make the loss surface between $\mu_{k-1}$ and $\mu_k$ smooth, and not to minimize the distance between $\mu_{k-1}$ and $\mu_k$. In convex optimization, the L2-regularizer makes the convex function strictly convex. Similarly, we hope L2-transfer with small $\lambda$ help to find a $\mu_k$ with a convex-like loss space between $\mu_{k-1}$ and $\mu_k$.

## 4.3 Drop-transfer

**Drop-transfer** is a novel method devised in this paper. Drop-transfer is a variant of dropout where $\mu_{k-1}$ is the zero point of the dropout procedure. In the training phase, the following $\hat{\mu}_{k,i}$ is used for the weight vector corresponding to the $i$th node $\mu_{k,i}$:

$$\hat{\mu}_{k,i} = \begin{cases} \mu_{k-1,i}, & \text{if } i\text{th node is turned off} \\ \frac{1}{1-p} \cdot \mu_{k,i} - \frac{p}{1-p} \cdot \mu_{k-1,i}, & \text{otherwise} \end{cases} \tag{11}$$

where $p$ is the dropout ratio. Notice that the expectation of $\hat{\mu}_{k,i}$ is $\mu_{k,i}$.

Table 1: The averaged accuracies on the disjoint MNIST for two sequential tasks (Top) and the shuffled MNIST for three sequential tasks (Bottom). The untuned setting refers to the most natural hyperparameter in the equation of each algorithm, whereas the tuned setting refers to using heuristic hand-tuned hyperparameters. Hyperparam denotes the main hyperparameter of each algorithm. For IMM with transfer, only $\alpha$ is tuned. The numbers in the parentheses refer to standard deviation. Every IMM uses weight-transfer.

| | Explanation of | Untuned | | Tuned | |
|---|---|---|---|---|---|
| **Disjoint MNIST Experiment** | Hyperparam | Hyperparam | Accuracy | Hyperparam | Accuracy |
| SGD [3] | epoch per dataset | 10 | 47.72 ($\pm$ 0.11) | 0.05 | 71.32 ($\pm$ 1.54) |
| L2-transfer [25] | $\lambda$ in (10) | - | - | 0.05 | 85.81 ($\pm$ 0.52) |
| **Drop-transfer** | $p$ in (11) | 0.5 | 51.72 ($\pm$ 0.79) | 0.5 | 51.72 ($\pm$ 0.79) |
| EWC [8] | $\lambda$ in (20) | 1.0 | 47.84 ($\pm$ 0.04) | 600M | 52.72 ($\pm$ 1.36) |
| **Mean-IMM** | $\alpha_2$ in (4) | 0.50 | 90.45 ($\pm$ 2.24) | 0.55 | 91.92 ($\pm$ 0.98) |
| **Mode-IMM** | $\alpha_2$ in (7) | 0.50 | 91.49 ($\pm$ 0.98) | 0.45 | 92.02 ($\pm$ 0.73) |
| **L2-transfer + Mean-IMM** | $\lambda / \alpha_2$ | 0.001 / 0.50 | 78.34 ($\pm$ 1.82) | 0.001 / 0.60 | 92.62 ($\pm$ 0.95) |
| **L2-transfer + Mode-IMM** | $\lambda / \alpha_2$ | 0.001 / 0.50 | 92.52 ($\pm$ 0.41) | 0.001 / 0.45 | 92.73 ($\pm$ 0.35) |
| **Drop-transfer + Mean-IMM** | $p / \alpha_2$ | 0.5 / 0.50 | 80.75 ($\pm$ 1.28) | 0.5 / 0.60 | 92.64 ($\pm$ 0.60) |
| **Drop-transfer + Mode-IMM** | $p / \alpha_2$ | 0.5 / 0.50 | 93.35 ($\pm$ 0.49) | 0.5 / 0.50 | 93.35 ($\pm$ 0.49) |
| **L2, Drop-transfer + Mean-IMM** | $\lambda / p / \alpha_2$ | 0.001 / 0.5 / 0.50 | 66.10 ($\pm$ 3.19) | 0.001 / 0.5 / 0.75 | **93.97** ($\pm$ 0.23) |
| **L2, Drop-transfer + Mode-IMM** | $\lambda / p / \alpha_2$ | 0.001 / 0.5 / 0.50 | **93.97** ($\pm$ 0.32) | 0.001 / 0.5 / 0.45 | **94.12** ($\pm$ 0.27) |
| | | | | | |
| **Shuffled MNIST Experiment** | | Hyperparam | Accuracy | Hyperparam | Accuracy |
| SGD [3] | epoch per dataset | 60 | 89.15 ($\pm$ 2.34) | - | $\sim$95.5 [8] |
| L2-transfer [25] | $\lambda$ in (10) | - | - | 1e-3 | 96.37 ($\pm$ 0.62) |
| **Drop-transfer** | $p$ in (11) | 0.5 | 94.75 ($\pm$ 0.62) | 0.2 | 96.86 ($\pm$ 0.21) |
| EWC [8] | $\lambda$ in (20) | - | - | - | $\sim$**98.2** [8] |
| **Mean-IMM** | $\alpha_3$ in (4) | 0.33 | 93.23 ($\pm$ 1.37) | 0.55 | 95.02 ($\pm$ 0.42) |
| **Mode-IMM** | $\alpha_3$ in (7) | 0.33 | 98.02 ($\pm$ 0.05) | 0.60 | 98.08 ($\pm$ 0.08) |
| **L2-transfer + Mean-IMM** | $\lambda / \alpha_3$ | 1e-4 / 0.33 | 90.38 ($\pm$ 1.74) | 1e-4 / 0.65 | 95.93 ($\pm$ 0.31) |
| **L2-transfer + Mode-IMM** | $\lambda / \alpha_3$ | 1e-4 / 0.33 | **98.16** ($\pm$ **0.08**) | 1e-4 / 0.60 | **98.30** ($\pm$ **0.08**) |
| **Drop-transfer + Mean-IMM** | $p / \alpha_3$ | 0.5 / 0.33 | 90.79 ($\pm$ 1.30) | 0.5 / 0.65 | 96.49 ($\pm$ 0.44) |
| **Drop-transfer + Mode-IMM** | $p / \alpha_3$ | 0.5 / 0.33 | 97.80 ($\pm$ 0.07) | 0.5 / 0.55 | 97.95 ($\pm$ 0.08) |
| **L2, Drop-transfer + Mean-IMM** | $\lambda / p / \alpha_3$ | 1e-4 / 0.5 / 0.33 | 89.51 ($\pm$ 2.85) | 1e-4 / 0.5 / 0.90 | 97.36 ($\pm$ 0.19) |
| **L2, Drop-transfer + Mode-IMM** | $\lambda / p / \alpha_3$ | 1e-4 / 0.5 / 0.33 | 97.83 ($\pm$ 0.10) | 1e-4 / 0.5 / 0.50 | 97.92 ($\pm$ 0.05) |

There are studies [27, 20] that have interpreted dropout as an exponential ensemble of weak learners. By this perspective, since the marginalization of output distribution over the whole weak learner is intractable, the parameters multiplied by the inverse of the dropout rate are used for the test phase in the procedure. In other words, the parameters of the weak learners are, in effect, simply averaged oversampled learners by dropout. At the process of drop-transfer in our continual learning setting, we hypothesize that the dropout process makes the averaged point of two arbitrary sampled points using Equation 11 a good estimator.

We investigated the search space of the loss function of the MLP trained from the MNIST handwritten digit recognition dataset for with and without dropout regularization, to supplement the evidence of the described hypothesis. Dropout regularization makes the accuracy of a sampled point from dropout distribution and an average point of two sampled parameters, from 0.450 ($\pm$ 0.084) to 0.950 ($\pm$ 0.009) and 0.757 ($\pm$ 0.065) to 0.974 ($\pm$ 0.003), respectively. For the case of both with and without dropout, the space between two arbitrary samples is empirically convex, and fits to the second-order equation. Based on this experiment, we expect not only that the search space of the loss function between modern neural networks can be easily nearly convex [23], but also that regularizers, such as dropout, make the search space smooth and the point in the search space have a good accuracy in continual learning.

## 5 Experimental Results

We evaluate our approach on four experiments, whose settings are intensively used in the previous works [4, 8, 7, 12]. For more details and experimental results, see Appendix D. The source code for the experiments is available in Github repository[1].

**Disjoint MNIST Experiment.** The first experiment is the disjoint MNIST experiment [4]. In this experiment, the MNIST dataset is divided into two datasets: the first dataset consists of only digits {0, 1, 2, 3, 4} and the second dataset consists of the remaining digits {5, 6, 7, 8, 9}. Our task is 10-

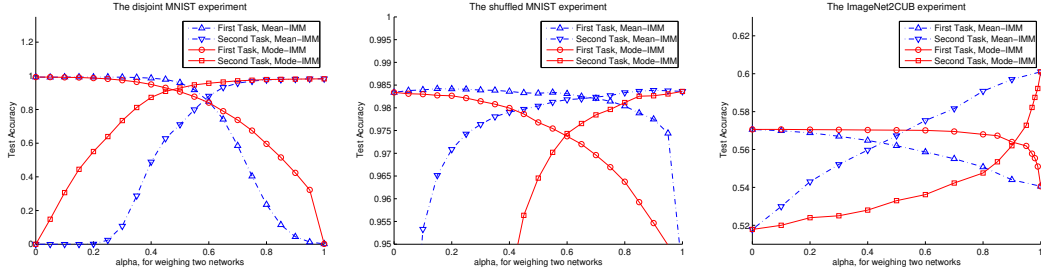

Figure 3: Test accuracies of two IMM models with weight-transfer on the disjoint MNIST (Left), the shuffled MNIST (Middle), and the ImageNet2CUB experiment (Right). $\alpha$ is a hyperparameter that balances the information between the old and the new task.

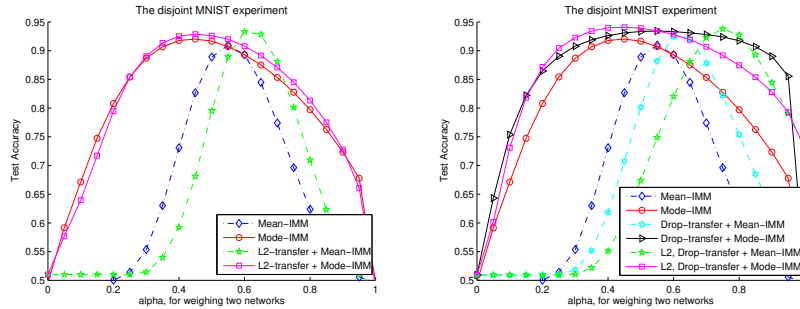

Figure 4: Test accuracies of IMM with various transfer techniques on the disjoint MNIST. Both L2-transfer and drop-transfer boost the performance of IMM and make the optimal value of $\alpha$ larger than 1/2. However, drop-transfer tends to make the accuracy curve more smooth than L2-transfer does.

class joint categorization, unlike the setting in the previous work, which considers two independent tasks of 5-class categorization. Because the inference should decide whether a new instance comes from the first or the second task, our task is more difficult than the task of the previous work.

We evaluate the models both on the untuned setting and the tuned setting. The untuned setting refers to the most natural hyperparameter in the equation of each algorithm. The tuned setting refers to using heuristic hand-tuned hyperparameters. Consider that tuned hyperparameter setting is often used in previous works of continual learning as it is difficult to define a validation set in their setting. For example, when the model needs to learn from the new task after learning from the old task, a low learning rate or early stopping without a validation set, or arbitrary hyperparameter for balancing is used [3, 8]. We discover hyperparameters in the tuned setting not only to find the oracle performance of each algorithm, but also to show that there exist some paths consisting of the point that performs reasonably for both tasks. Hyperparam in Table 1 denotes hyperparameter mainly searched in the tuned setting. Table 1 (Top) and Figure 3 (Left) shows the experimental results from the disjoint MNIST experiment.

In our experimental setting, the usual SGD-based optimizers always perform less than 50%, because the biases of the output layer for the old task are always pushed to large negative values, which implies that our task is difficult. Figure 4 also shows that mode-IMM is robust with $\alpha$ and the optimal $\alpha$ of mean-IMM is larger than $1/2$ in the disjoint MNIST experiment.

**Shuffled MNIST Experiment.** The second experiment is the shuffled MNIST experiment [3, 8] of three sequential tasks. In this experiment, the first dataset is the same as the original MNIST dataset. However, in the second dataset, the input pixels of all images are shuffled with a fixed, random permutation. In previous work, EWC reaches the performance level of the batch learner, and it is argued that EWC overcomes catastrophic forgetting in some domains. The experimental details are similar to the disjoint MNIST experiment, except all models are allowed to use dropout regularization. In the experiment, the first dataset is the same as the original MNIST dataset. However, in the second and the third dataset, the input pixels of all images are shuffled with a fixed, random permutation,

Table 2: Experimental results on the Lifelog dataset among different classes (location, sub-location, and activity) and different subjects (A, B, C). Every IMM uses weight-transfer.

|  | Location | Sub-location | Activity | A | B | C |
|---|---|---|---|---|---|---|
| Dual memory architecture [12] | **78.11** | 72.36 | 52.92 | 67.02 | 58.80 | 77.57 |
| **Mean-IMM** | 77.60 | 73.78 | 52.74 | 67.03 | 57.73 | **79.35** |
| **Mode-IMM** | 77.14 | **75.76** | **54.07** | **67.97** | **60.12** | 78.89 |

respectively. Therefore, the difficulty of the three datasets is the same, though a different solution is required for each dataset.

Table 1 (Bottom) and Figure 3 (Middle) shows the experimental results from the shuffled MNIST experiment. Notice that accuracy of drop-transfer ($p = 0.2$) alone is 96.86 ($\pm$ 0.21) and L2-transfer ($\lambda$ = 1e-4) + drop-transfer ($p = 0.4$) alone is 97.61 ($\pm$ 0.15). These results are competitive to EWC without dropout, whose performance is around 97.0.

**ImageNet to CUB Dataset.** The third experiment is the ImageNet2CUB experiment [7], the continual learning problem from the ImageNet dataset to the Caltech-UCSD Birds-200-2011 fine-grained classification (CUB) dataset [28]. The numbers of classes of ImageNet and CUB dataset are around 1K and 200, and the numbers of training instances are 1M and 5K, respectively. In the ImageNet2CUB experiment, the last-layer is separated for the ImageNet and the CUB task. The structure of AlexNet is used for the trained model of ImageNet [29]. In our experiment, we match the moments of the last-layer fine-tuning model and the LwF model, with mean-IMM and mode-IMM.

Figure 3 (Right) shows that mean-IMM moderately balances the performance of two tasks between two networks. However, the balanced hyperparameter of mode-IMM is far from $\alpha = 0.5$. We think that it is because the scale of the Fisher matrix $F$ is different between the ImageNet and the CUB task. Since the number of training data of the two tasks is different, the mean of the square of the gradient, which is the definition of $F$, tends to be different. This implies that the assumption of mode-IMM does not always hold for heterogeneous tasks. See Appendix D.3 for more information including the learning methods of IMM where a different class output layer or a different scale of the dataset is used.

Our results of IMM with LwF exceed the previous state-of-the-art performance, whose model is also LwF. This is because, in the previous works, the LwF model is initialized by the last-layer fine-tuning model, not directly by the original AlexNet. In this case, the performance loss of the old task is not only decreased, but also the performance gain of the new task is decreased. The accuracies of our mean-IMM ($\alpha = 0.5$) are 56.20 and 56.73 for the ImageNet task and the CUB task, respectively. The gains compared to the previous state-of-the-art are +1.13 and -1.14. In the case of mean-IMM ($\alpha = 0.8$) and mode-IMM ($\alpha = 0.99$), the accuracies are 55.08 and 59.08 (+0.01, +1.12), and 55.10 and 59.12 (+0.02, +1.35), respectively.

**Lifelog Dataset.** Lastly, we evaluate the proposed methods on the Lifelog dataset [12]. The Lifelog dataset consists of 660,000 instances of egocentric video stream data, collected over 46 days from three participants using Google Glass [30]. Three class categories, location, sub-location, and activity, are labeled on each frame of video. In the Lifelog dataset, the class distribution changes continuously and new classes appear as the day passes. Table 2 shows that mean-IMM and mode-IMM are competitive to the dual-memory architecture, the previous state-of-the-art ensemble model, even though IMM uses single network.

## 6 Discussion

**A Shift of Optimal Hyperparameter of IMM.** The tuned setting shows there often exists some $\alpha$ which makes the performance of the mean-IMM close to the mode-IMM. However, in the untuned hyperparameter setting, mean-IMM performs worse when more transfer techniques are applied. Our Bayesian interpretation in IMM assumes that the SGD training of the $k$-th network $\mu_k$ is mainly affected by the $k$-th task and is rarely affected by the information of the previous tasks. However, transfer techniques break this assumption; thus the optimal $\alpha$ is shifted to larger than $1/k$. Fortunately, mode-IMM works more robustly than mean-IMM where transfer techniques are applied.

Figure 4 illustrates the change of the test accuracy curve corresponding to the applied transfer techniques and the following shift of the optimal $\alpha$ in mean-IMM and mode-IMM.

**Bayesian Approach on Continual Learning.** Kirkpatrick et al. [8] interpreted that the Fisher matrix $F$ as weight importance in explaining their EWC model. In the shuffled MNIST experiment, since a large number of pixels always have a value of zero, the corresponding elements of the Fisher matrix are also zero. Therefore, EWC does work by allowing weights to change, which are not used in the previous tasks. On the other hand, mode-IMM also works by selectively balancing between two weights using variance information. However, these assumptions on weight importance do not always hold, especially in the disjoint MNIST experiment. The most important weight in the disjoint MNIST experiment is the bias term in the output layer. Nevertheless, these bias parts of the Fisher matrix are not guaranteed to be the highest value nor can they be used to balance the class distribution between the first and second task. We believe that using only the diagonal of the covariance matrix in Bayesian neural networks is too naïve in general and that this is why EWC failed in the disjoint MNIST experiment. We think it could be alleviated in future work by using a more complex prior, such as a matrix Gaussian distribution considering the correlations between nodes in the network [31].

**Balancing the Information of an Old and a New Task.** The IMM procedure produces a neural network without a performance loss for $k$th task $\mu_k$, which is better than the final solution $\mu_{1:k}$ in terms of the performance of the $k$th task. Furthermore, IMM can easily weigh the importance of tasks in IMM models in real time. For example, $\alpha_t$ can be easily changed for the solution of mean-IMM $\mu_{1:k} = \sum_t^k \alpha_t \mu_t$. In actual service situations of IT companies, the importance of the old and the new task frequently changes in real time, and IMM can handle this problem. This property differentiates IMM from the other continual learning methods using the regularization approach, including LwF and EWC.

# 7  Conclusion

Our contributions are four folds. First, we applied mean-IMM to the continual learning of modern deep neural networks. Mean-IMM makes competitive results to comparative models and balances the information between an old and a new network. We also interpreted the success of IMM by the Bayesian framework with Gaussian posterior. Second, we extended mean-IMM to mode-IMM with the interpretation of mode-finding in the mixture of Gaussian posterior. Mode-IMM outperforms mean-IMM and comparative models in various datasets. Third, we introduced drop-transfer, a novel method proposed in the paper. Experimental results showed that drop-transfer alone performs well and is similar to the EWC without dropout, in the domain where EWC rarely forgets. Fourth, We applied various transfer techniques in the IMM procedure to make our assumption of Gaussian distribution reasonable. We argued that not only the search space of the loss function among neural networks can easily be nearly convex, but also regularizers, such as dropout, make the search space smooth, and the point in the search space have good accuracy. Experimental results showed that applying transfer techniques often boost the performance of IMM. Overall, we made state-of-the-art performance in various datasets of continual learning and explored geometrical properties and a Bayesian perspective of deep neural networks.

# Acknowledgments

The authors would like to thank Jiseob Kim, Min-Oh Heo, Donghyun Kwak, Insu Jeon, Christina Baek, and Heidi Tessmer for helpful comments and editing. This work was supported by the Naver Corp. and partly by the Korean government (IITP-R0126-16-1072-SW.StarLab, IITP-2017-0-01772-VTT, KEIT-10044009-HRI.MESSI, KEIT-10060086-RISF). Byoung-Tak Zhang is the corresponding author.

## Footnotes

[1]https://github.com/btjhjeon/IMM_tensorflow

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
