[Supplementary Material · NIPS17_v28_3_appendix.pdf]

## APPENDIX A. Modes in the Mixture of Gaussian

According to Ray and Lindsay [21], all the critical points $\theta$ of a mixture of Gaussian (MoG) with two components are in one curve as the following equation with $0 < \alpha < 1$.

$$\theta = ((1 - \alpha)\Sigma_1^{-1} + \alpha\Sigma_2^{-1})^{-1}((1 - \alpha)\Sigma_1^{-1}\mu_1 + \alpha\Sigma_2^{-1}\mu_2) \tag{12}$$

The proof is as follows. Imagine two Gaussian distribution $q_1$ and $q_2$, such as in Equation 2.

$$q_1 \equiv q_1(\theta; \mu_1, \Sigma_1) = \frac{1}{\sqrt{(2\pi)^D|\Sigma_1|}} \exp\left(-\frac{1}{2}(\theta - \mu_1)^T\Sigma_1^{-1}(\theta - \mu_1)\right) \tag{13}$$

$$q_2 \equiv q_2(\theta; \mu_2, \Sigma_2) = \frac{1}{\sqrt{(2\pi)^D|\Sigma_2|}} \exp\left(-\frac{1}{2}(\theta - \mu_2)^T\Sigma_1^{-1}(\theta - \mu_2)\right) \tag{14}$$

$D$ is the dimension of the Gaussian distribution. Mixture of two Gaussian $q_1$ and $q_2$ with the equal mixing ratio (i.e., 1:1) is $q_1/2 + q_2/2$. The derivation of the MoG is as follows:

$$\frac{\partial(q_1/2 + q_2/2)}{\partial\theta} = -\frac{q_1}{2}(\Sigma_1^{-1}(\theta - \mu_1)) - \frac{q_2}{2}(\Sigma_2^{-1}(\theta - \mu_2)) = 0 \tag{15}$$

If we set Equation 15 to 0, to find all critical points, the following equation holds:

$$\theta = (q_1\Sigma_1^{-1} + q_2\Sigma_2^{-1})^{-1}(q_1\Sigma_1^{-1}\mu_1 + q_2\Sigma_2^{-1}\mu_2) \tag{16}$$

When $\alpha$ is set to $\frac{q_2}{q_1+q_2}$, Equation 12 holds.

Note that $\alpha_k$ is a function of $\theta$, so $\theta$ cannot be calculated in a closed-form from Equation 16. However, the optimal $\theta$ is in the set $\{\theta | \theta = ((1 - \alpha)\Sigma_1^{-1} + \alpha\Sigma_2^{-1})^{-1}((1 - \alpha)\Sigma_1^{-1}\mu_1 + \alpha\Sigma_2^{-1}\mu_2), 0 < \alpha < 1\}$, which motivates our mode-IMM method.

In our study IMM uses diagonal covariance matrices, which means that there is no correlation between parameters. This diagonal assumption is useful, since it decreases the number of parameters for each covariance matrix from $O(D^2)$ to $O(D)$. Based on this, the $\theta$ in Equation 12 is defined as follows:

$$\theta_v = \frac{(1 - \alpha) \cdot \mu_{1,v}/\sigma_{1,v}^2 + \alpha \cdot \mu_{2,v}/\sigma_{2,v}^2}{(1 - \alpha)/\sigma_{1,v}^2 + \alpha/\sigma_{2,v}^2} \tag{17}$$

$v$ denotes an index of the parameter vector. $\mu_{\cdot,v}$ and $\sigma_{\cdot,v}^2$ are scalar.

For MoG with two components in $K$ dimension, the number of modes can be at most $K + 1$ [32]. Therefore, it is hard to find all modes in high-dimensional Gaussian in general.

The property of critical points of a MoG with two components can be extended to the case of $K$ components. The following equation holds:

$$\theta = (\sum_{k=1}^{K} \alpha_k\Sigma_k^{-1})^{-1}(\sum_{k=1}^{K} \alpha_k\Sigma_k^{-1}\mu_k), \tag{18}$$

where $0 < \alpha_k < 1$ for all $k$ and $\sum_k \alpha_k = 1$. There is no tight upper bound on the number of modes of MoG in general. There is a guess that, for all $D, K \geq 1$, the upper bound is $_{(D+K-1)}C_D$ [33].

## APPENDIX B. Bayesian Neural Networks and Continual Learning

**Bayesian Neural Networks.** Bayesian neural networks (BNN) assume an uncertainty for the whole parameter in neural networks so that the posterior distribution can be obtained [10]. Previous studies

---
**Algorithm 1** IMM with weight-transfer, L2-transfer
---
**Input:** data $\{(X_1, y_1),...,(X_K, y_K)\}$, balancing hyperparameter $\alpha$
**Output:** $w_{1:K}$
$w_0 \leftarrow$ InitializeNN()
**for** $k = 1{:}K$ **do**
   $w_{k*} \leftarrow w_{k-1}$
   Train$(w_{k*}, X_k, y_k)$ with $L(w_{k*}, X_k, y_k) + \lambda \cdot ||w_{k*} - w_{k-1}||_2^2$
   **if** type is mean-IMM **then**
      $w_{1:k} \leftarrow \sum_t^k \alpha_t w_{t*}$
   **else if** type is mode-IMM **then**
      $F_{k*} \leftarrow$ CalculateFisherMatrix$(w_{k*}, X_k, y_k)$
      $\Sigma_{1:k} \leftarrow (\sum_t^k \alpha_t F_{t*})^{-1}$
      $w_{1:k} \leftarrow \Sigma_{1:k} \cdot (\sum_t^k \alpha_t F_{t*} w_{t*})$
   **end if**
**end for**
---

have argued that BNN regularizes better than NN, and provides a confidence interval for the output estimation of each input instance. Current research on BNN, to the best of our knowledge, uses Gaussian distributions as the posteriors of the parameters. In the Gaussian assumption, because tracking the entire information of a covariance matrix is too expensive, researchers usually use only the diagonal term for the covariance matrix, where the posterior distribution is fully factorized for each parameter. However, the methods using full covariance were also suggested recently [31]. To estimate a covariance matrix most studies use stochastic gradient variational Bayes (SGVB), where a sampled point from the posterior distribution by Monte Carlo is used in the training phases [34]. Alternatively, Kirkpatrick et al. [8] approximated the covariance matrix as an inverse of a Fisher matrix. This approximation makes the computational cost of the inference of a covariance matrix cheaper when the update of covariance information is not needed in the training phase. Our method follows the approach using the Fisher matrix.

**Elastic Weight Consolidation.** We compare the work of Kirkpatrick et al. [8] to the results of our framework. The mechanism of EWC follows sequential Bayesian estimation. EWC maximizes the following terms by gradient descent to get the solution $\mu_{1:K}$.

$$
\begin{aligned}
\log p_{1:K} &\approx \log p(y_K | X_K, \theta) + \lambda \cdot \log p_{1:(K-1)} + C \\
&\approx \log p(y_K | X_K, \theta) + \lambda \cdot \sum_{k=1}^{K-1} \log q_{1:k} + C \\
&= \log p(y_K | X_K, \theta) - \frac{\lambda}{2} \cdot \sum_{k=1}^{K-1} (\theta - \mu_{1:k})^T \tilde{\Sigma}_k^{-1} (\theta - \mu_{1:k}) + C'
\end{aligned}
\tag{19}
$$

$p_k$ is empirical posterior distribution of $k$th task, and $q_k \sim N(\mu_k, \Sigma_k)$ is an approximation of $p_k$. In EWC, $\tilde{\Sigma}_k^{-1}$ is also approximated by the diagonal term of Fisher matrix $\tilde{F}_k$ with respect to $\mu_{1:k}$ and $X_k$.

When moving to a third task, EWC uses the penalty term of both first and second network (i.e., $\mu_1$ and $\mu_{1:2}$). Although this heuristic works reasonably in the experiments in their paper, it does not match to the philosophy of Bayesian.

**Learning without Forgetting.** We compare the work of Li and Hoiem [7]. Although LwF does not explicitly assume Bayesian, the approach can be represented nonetheless as follows:

$$
\log p_{1:K} \approx \log p(y_K | X_K, \theta) + \lambda \cdot \sum_{k=1}^{K-1} \log p(\hat{y}_k | X_K, \theta)
\tag{20}
$$

Where $\hat{y_k}$ is the output from $\mu_k$ with input $X_K$. This framework is promising where the properties of a pseudo training set of $k$th task $(X_K, \hat{y}_k)$ is similar to the ideal training set $(X_k, y_k)$.

## APPENDIX C. Example Algorithms of Incremental Moment Matching

Two moment matching methods: mean-IMM and mode-IMM, and three transfer learning techniques: weight-transfer, L2-transfer, and drop-transfer, are combined to make various continual learning algorithms in our study. Algorithm 1 describes mean-IMM and mode-IMM with weight-transfer and L2-transfer.

## APPENDIX D. Experimental Details

Appendix D further explains following issues, 1) additional explanation of the untuned setting and tuned setting 2) techniques for IMM with a different class output layer for each task 3) other experimental details.

### D.1 Disjoint MNIST Experiment

We first explain the untuned setting and the tuned setting in detail. The untuned setting refers to the most natural hyperparameter in the equation of each algorithm, whereas the tuned setting refers to using heuristic hand-tuned hyperparameters. For mean-IMM, it is most natural to evenly average K models and $1/K$ is the most natural $\alpha_k$ value for $K$ sequential tasks. For EWC, 1 is the most natural $\lambda$ value in Equation 19, because EWC is derived from the equation of sequential Bayesian. For L2-transfer, there is no natural hyperparameter value in Equation 10, so we need to heuristically choose a $\lambda$ value, which is not too small but does not damage the performance of the new network for the new task.

In the SGD, the number of epochs for the dataset (epoch per dataset) for the second task corresponds to the hyperparameter. The unit is how much of the network is trained from the whole data at once. In the L2-transfer and EWC, $\lambda$ in Equations 10 and 19 corresponds to their hyperparameter. In the mean-IMM and mode-IMM, $\alpha_K$ in Equations 4 and 7 corresponds to the hyperparameter. In the drop-transfer, dropout ratio $p$ in Equation 11 corresponds to the hyperparameter.

All of the explained hyperparameters are devised to balance the information between the old and new tasks. If $\lambda/(1 + \lambda) = 1$ or $\alpha_1 = 1$, the final network of the algorithms is the same as the network for the first task. If $1/(1 + \lambda) = 1$ or $\alpha_K = 1$, the final network is the same as the network for the last task.

We used multi-layer perceptrons (MLP) with [784-800-800-10] as the number of nodes, ReLU as the activation function, and vanilla SGD as the optimizer for all of the experiments. We set the epoch per dataset to 10, unless otherwise noted. The entire IMM model uses weight-transfer to smooth the loss surface of the model. Without weight-transfer, our IMM model does not work at all. In our experiments, all models only use one 10-way softmax output layer. For only SGD, dropout is used as proposed in Goodfellow et al. [3], but dropout does not help much.

Each accuracy was measured by averaging the results of 10 experiments. In the experiment, IMM outperforms comparative models by a significant margin. In the tuned experiment, the performance of the IMM models exceeds 90%, and the performance increases more when more transfer techniques are applied. Among all the models, weight-transfer + L2-transfer + drop-transfer + mode-IMM performs the best and its performance is greater than 94%. However, the comparative models fail to reach greater than 90%. Existing regularizer including dropout does not improve the comparative models.

### D.2 Shuffled MNIST Experiment

The second experiment is the shuffled MNIST experiment for three sequential tasks. For the hyperparameter of IMM, we set $\alpha_1$ and $\alpha_2$ as the same value, and tune only $\alpha_3$. Table 1 (Bottom) shows the experimental results. The performance of SGD + dropout and EWC + dropout comes from the report in [8]. Changing only the epoch does not significantly increase the performance in SGD. The results show that our IMM paradigm performs similarly to EWC in a case where EWC performs well. Dropout regularization in the task makes both our models and comparative models perform better.

Figure 5: (Left) Illustration of the effect of the strategy of re-weighing on the new last-layer. Mode-IMM refers to the original mode-IMM devised for the ImageNet2CUB experiments. In naïve mode-IMM, the second last-layer of the second network is used for the second last-layer of the final IMM model. (Right) The results of mode-IMM with changing the balancing hyperparameter $\alpha$ to the re-scaled balancing hayperparameter $\hat{\alpha}$ with the scale of the Fisher matrix of each network.

In our IMM framework, weight-transfer, L2-transfer, and drop-transfer all take $\mu_{k-1}$ as the reference models of the transfer for training $\mu_k$. In other words, weight-transfer initializes $\mu_k$ with $\mu_{k-1}$, L2-transfer uses a regularization term to minimize the Euclidean distance between $\mu_{k-1}$ and $\mu_k$, and drop-transfer uses a $\mu_{k-1}$ as the zero point of the dropout procedure. All three transfer techniques can be considered to change the reference point to, for example, $\mu_{1:(k-1)}^{mean}$ or $\mu_{1:(k-1)}^{mode}$, as previous works do [8]. However, all these alternatives make performances worse in the shuffled MNIST experiment. We argued that our utilization of transfer techniques is devised not to minimize the distance between $\mu_{k-1}$ and $\mu_k$, but to help find a $\mu_k$ with a smooth and convex-like loss space between $\mu_{k-1}$ and $\mu_k$.

### D.3 ImageNet to Other Image Datasets

When each task needs a different class output layer, IMM requires additional techniques. There is no counterpart weight matrix in the last-layer of the first network representing the second task, nor the second last-layer of the first network. To tackle this problem, we add the training process of the last-layer fine-tuning model to the IMM procedure; we match the moments of the last-layer fine-tuning model with the original new network for the new task. Last-layer fine-tuning is the model the last-layer is only fine-tuned for each new task; thus it does not make a performance loss for the first task, but does not often learn enough for new tasks.

The technique utilizing the last-layer fine-tuning model makes mean-IMM work in the case of different class output layers, but it is not enough for mode-IMM. It is not possible to calculate a proper Fisher matrix of the second last-layer in the first network for the first dataset. As the Fisher matrix is defined with the gradient from the loss of the first task, elements of the Fisher matrix have a value of zero. However, a zero matrix not only is what we do not want but also degenerates the performance of mode-IMM. To tackle this problem, we apply mean-IMM for the last-layer with a re-scaling. We change the mixing ratios $\alpha_1 : \alpha_2$ to $\hat{\alpha}_1 : \hat{\alpha}_2 = \alpha_1 : \alpha_2 \cdot \frac{|\hat{w}_1|}{|\hat{w}_1| + |\hat{w}_2|}$ for the re-scaling, where $|\hat{w}_1|$ and $|\hat{w}_2|$ is the average of the whole element of weight matrix in the layer before the last-layer, in the first and the second task.

In our ImageNet2CUB experiment, the moments of the last-layer fine-tuning model and the LwF model are matched. Though LwF does not perform well in our previous experiments, it is known that LwF performs well when the size of a new dataset is small relative to the old dataset, as in the ImageNet2CUB experiment.

Figure 5 (Left) compares the performances of mode-IMM models with different assumptions on the Fisher matrix. In naïve mode-IMM, the Fisher matrix of the second last-layer of the first network is a zero matrix. In other words, the second last-layer of the final naïve mode-IMM is the second last-layer of the second network. Naïve mode-IMM does not yield a good performance as we expect.

Table 3: Experimental results on the Lifelog dataset. Mean-IMM uses weight-transfer. Classification accuracies among different classes (Top) and different subjects (Bottom). In the experiment, our IMM paradigm achieves competitive results with the approach using an ensemble network, without additional cost for inference and learning.

| Algorithm | Location | Sub-location | Activity |
|---|---|---|---|
| Dual memory architecture [12] | **78.11** | 72.36 | 52.92 |
| **Mean-IMM** | 77.60 | 73.78 | 52.74 |
| **Mode-IMM** | 77.14 | **75.76** | **54.07** |
| Online fine-tuning | 68.27 | 64.13 | 50.00 |
| Last-layer fine-tuning | 74.58 | 69.30 | 52.22 |
| Naïve incremental bagging | 74.48 | 67.18 | 47.92 |
| Incremental bagging w/ transfer | 74.95 | 68.53 | 49.66 |

| Algorithm | A | B | C |
|---|---|---|---|
| Dual memory architecture [12] | 67.02 | 58.80 | 77.57 |
| **Mean-IMM** | 67.03 | 57.73 | **79.35** |
| **Mode-IMM** | **67.97** | **60.12** | 78.89 |
| Online fine-tuning | 53.01 | 56.54 | 72.85 |
| Last-layer fine-tuning | 63.31 | 55.83 | 76.97 |
| Naïve incremental bagging | 62.24 | 53.57 | 73.77 |
| Incremental bagging w/ transfer | 61.21 | 56.71 | 75.23 |

In Figure 5, scaled mode-IMM denotes the results of mode-IMM re-plotted by the $\hat{\alpha}$ as we defined above. The result shows that re-scaled mode-IMM performs similarly to mean-IMM in the ImageNet2CUB experiment.

**D.4 Lifelog Dataset**

The Lifelog dataset is the dataset recorded from Google Glass over 46 days from three participants. The 660,000 seconds of the egocentric video stream data reflects the behaviors of the participants. The dataset consists of 10 days of training data and 4 days of test data in order of time for each participant respectively. In the framework of Lee et al. [12], the network can be updated every day, but a new network can be made for the 3rd, 7th, and 10th day, with training data of 3, 4, and 3 days, respectively. Following this framework, our network is made in the 3rd, 7th, and 10th day, and then merged to previously trained networks. Our IMM used AlexNet pretrained by the ImageNet dataset [29] as the initial network. The experimental results on the Lifelog dataset are in Table 3, where the performance of models is from Lee et al. [12] except IMM.