[Reviews · NeurIPS 2017]

Reviewer 1



The authors propose several methods for achieving continual learning by merging Gaussian posteriors of networks trained on several separate tasks: either take the mean or the mode. The techniques are simple but well motivated. Experiments suggest that these methods improve catastrophic forgetting in several tasks. The paper has a nice review of related work, and the derivations and justifications are good. I found the paper an interesting read. The experiments appear to be well done and sufficient detail is included that they should be reimplementable. I have one question concerning (7) and (8): practically, as there is a matrix inverse involved, did the authors do something the ensure numerical stability?

Reviewer 2



[UPDATE] After digging through the supplementary material I have managed to find at least a hint towards possible limitations of the method being properly acknowledged. Not including an objective evaluation of limitations is a flaw of this otherwise well written paper, especially when the method relies crucially on weight transfer (as the authors point out outside the main paper, i.e. supplementary text and rebuttal). However, weight transfer is known to be an inadequate initialization technique between different problem classes and the authors don't clearly address this issue, nor do they properly qualify the applicability of the method. In balance, this paper does give sufficient evidence that weight transfer and some form of parameter averaging are promising directions of future investigation, at least in a subset of interesting cases. Perhaps with publication chances of expanding this subset increase. [REVIEW TEXT] The paper evaluates the idea of incremental moment matching (IMM) as a solution for catastrophic forgetting in neural networks. The method is thoroughly benchmarked, in several incarnations, against state-of-the-art baselines on standard ‘toy’ problems defined on top of MNIST, as well as more challenging ImagNet2CUB and the Lifelog dataset. A new parameterization, dubbed ‘drop-transfer’ is proposed as an alternative to standard weight initialization of model parameters on new tasks. Pros: - Comprehensive experiments are provided which show superior performance on standard MNIST classification variants, in comparison to previous methods. - Encouraging results are shown on natural image datasets, although here many earlier methods were not applied, so there were few literature results to compare against. Cons: - Although differences exist between the pairs or triplets of problems considered to illustrate ‘catastrophic forgetting’, technically speaking, these problems are extremely related, sharing either the input support or the class distribution to some degree; this is in contrast to current challenges in the field, which consider sequences of arbitrarily different reinforcement learning problems, such as different Atari video games (Kirkpatrick et al. 2016, Rusu et al., 2016). Therefore, I find it difficult to tell how proficient the method is at reducing catastrophic forgetting in the general case. I recommend considering more standard challenges from the literature. - Another important consideration for such works is evaluating how much learning is possible with the given model when following the proposed solution. Since catastrophic forgetting can be alleviated by no further learning throughout most of the model, it follows that proficient learning of subsequent tasks, preferably superior to starting from scratch, is a requirement for any ‘catastrophic forgetting’ solution. As far as I can tell the proposed method does not offer a plausible answer if tasks are too different, or simply unrelated; yet these are the cases most likely to induce catastrophic forgetting. In fact, it is well understood that similar problems which share the input domain can easily alleviate catastrophic forgetting by distilling the source task network predictions in a joint model using target dataset inputs (Li et al., 2016, Learning without Forgetting). In summary, I believe variations of the proposed method may indeed help solve catastrophic forgetting in a more general setting, but I would like to kindly suggest evaluating it against current standing challenges in the literature.